# Applying OGC Sensor Web Enablement Standards to Develop a TDR Multi-Functional Measurement Model

**DOI:** 10.3390/s19194070

**Published:** 2019-09-20

**Authors:** Chih-Chung Chung, Chih-Yuan Huang, Chih-Ray Guan, Ji-Hao Jian

**Affiliations:** 1Dept. of Civil Engineering, National Central University, 300 Zhongda, Rd., Zhongli Dist., Taoyuan 320, Taiwan; harry4558719@gmail.com (C.-R.G.); apple900d@gmail.com (J.-H.J.); 2Center for Space and Remote Sensing Research, National Central University, 300 Zhongda, Rd., Zhongli Dist., Taoyuan 320, Taiwan; cyhuang@csrsr.ncu.edu.tw

**Keywords:** Time-domain reflectometry (TDR), interoperability, Sensor Observation Service (SOS)

## Abstract

Time-domain reflectometry (TDR) is considered as a passive monitoring technique which reveals multi-functions, such as water level, bridge scour, landslide, and suspended sediment concentration (SSC), based on a single TDR device via multiplexing and related algorithms. The current platform for revealing TDR analysis and interpreted observations, however, is complex to access, thus a coherent data model and format for TDR heterogeneous data exchange is useful and necessary. To enhance the interoperability of TDR information, this research aims at standardizing the TDR data based on the Open Geospatial Consortium (OGC) Sensor Web Enablement (SWE) standards. To be specific, this study proposes a TDR sensor description model and an observation model based on the Sensor Model Language (SensorML) and Observation and Measurement (O&M) standards. In addition, a middleware was developed to translate existing TDR information to a Sensor Observation Service (SOS) web service. Overall, by standardizing TDR data with the OGC SWE open standards, relevant information for disaster management can be effectively and efficiently integrated in an interoperable manner.

## 1. Introduction

The technology of Internet-connected networks has brought evolutional changes from academic research to daily life in recent decades. Data collection is automated and convenient now, and related management has a profound impact on monitoring in civil engineering and disaster prevention domains. Time-domain reflectometry (TDR) technology, as a one-dimension cable radar, is one of the passive monitoring techniques featured by interpreting the reflected electromagnetic (EM) wave due to the different environmental changes. TDR has been widely used in geotechnical, structural, and hydrological engineering monitoring, and even in agriculture [1,2]. The TDR monitoring applications include bridge pier scour [3,4,5,6], suspended sediment concentration (SSC) [7], water pressure or level [8,9,10], soil moisture content [11,12,13,14,15], soil electrical conductivity (EC) [16,17,18], slope sliding [19,20,21,22], and mine subsidence [23]. These references to case studies and specific challenges encountered in geotechnical and soil science applications of TDR demonstrate what kinds of engineering decisions have been or may be supported by TDR measurements. In addition, a TDR monitoring system not only contains the monitoring equipment at the site, but also cooperates with the data transmission module (such as Electronic Industry Association Recommended Standard 232 (RS232), General Packet Radio Service (GPRS), Wireless Local Area Networking (WLAN), Low Power, Wide Area (LPWA) networking (LoRaWAN), or Narrow Band-Internet of Things (NB-IoT)), which can automatically retrieve information from in-situ to a server [24]. This kind of real-time data is valuable for disaster early warning. Furthermore, the cause or the failure mechanisms of disasters can be discovered by analyzing the historical data. Based on the effective data interchange for aforementioned TDR implementations, previous works have put considerable effort into developing some early data interchange models for TDR and related instrument data [25,26]. The current TDR analysis with the related parameters, however, is still complex to access. Thus, well-considered models for interoperability are useful and necessary.

To improve the interoperability of TDR data, there are many feasible options from Internet of Things (IoT) and Sensor Web standard-definition bodies. As the IoT has received much attention recently, private companies have defined various proprietary data models and service protocols. This heterogeneity issue is usually solved with hub solutions that communicate with different ecosystems via customized connectors [27]. However, the hub approach, which is achieved by implementing different connectors corresponding to different Sensor Web and Internet of Things (SW-IoT) ecosystems, needs to use customized connectors for different protocols, leading to a significant development cost.

On the other hand, following open standards is an attractive approach to achieve data interoperability. Among the potential open-standard solutions for TDR data, the Open Geospatial Consortium (OGC) Sensor Web Enablement (SWE) provides a set of standards that are commonly seen as a comprehensive solution for the Sensor Web on the application layer [28]. For example, the Observation and Measurement (O&M) standard defines a general but complete data model for sensor observations [29], and the Sensor Model Language (SensorML) provides a flexible framework and schema to describe various types of sensor metadata based on an Extensible Markup Language (XML) derivative [30]. XML is one of the markup languages that defines rules for encoding documents in a format that is both human-readable and machine-readable. In addition, the Sensor Observation Service (SOS) standard is defined to provide a coherent service protocol for sensor data sharing [31].

Although the OGC SWE aims at providing an interoperable framework for general sensor data, specific domain data models within the SWE ecosystem still need to be defined to really achieve interoperability in each domain. For example, WaterML 2.0 is one of the standard information models for the representation of water observational data, which allows the exchange of such datasets across information systems [32]. Based on the existing OGC standards, it attempts being an interoperable exchange format that may be re-used to address a range of exchange requirements.

In addition, Sofos et al. [33] implemented the OGC O&M and SOS in describing the land surveying measurement model, especially focusing on the Total Station/Positioning System (TPS), and XML request documents were subsequently developed. Stasch et al. [34] developed coupling sensor observation services and web processing services for online geoprocessing in water dam monitoring by implementing the standardized XML-based interface and lightweight Representational State Transfer (REST) APIs, where REST is a software architectural style that defines constraints to be used for creating Web services. Web services that conform to the REST architectural style are called RESTful Web services (RWS) and provide interoperability between computer systems on the Internet [35]. Based on the aforementioned successful cases, this study aims at proposing a domain solution for integrating the TDR data with the OGC SWE ecosystem.

Furthermore, one thing should be mentioned is that the OGC SWE proposes a new service protocol standard, the SensorThings API [36], to share sensor data in a REST service. The SensorThings API applies the Java Script Object Notation (JSON) format to represent sensor data, where JSON is considered as a low-overhead alternative to XML [37]. Although SOS is based on Simple Object Access Protocol (SOAP) XML, which is heavier than the JSON format, XML can represent the basic and fundamental content by permitting strong typing, user-defined types, predefined tags, and a formal structure for describing the TDR model.

Overall, this study attempts to propose a recommended model for TDR monitoring data under the OGC SWE standards to improve data interoperability. Afterward, a case study is then introduced by integrating a TDR monitoring system with an SOS service. Then, an OGC Web Processing Service (WPS) service supporting landslide monitoring and pre-warning strategies is demonstrated to reveal an interoperable manner by using TDR.

## 2. TDR Basics 

TDR technology was developed from the detection of cable deformation and fault in the telecommunications industry in 1950. Today’s TDR technology is used widely in geotechnical engineering, agricultural engineering, environmental science, electronic engineering, and earth engineering [1]. The operating principle of TDR is similar to that of radar, except that the EM pulse travels through a cable rather than free space. A TDR device, as shown in Figure 1, includes an EM step-pulse generator, a signal sampler with or without an oscilloscope. The step-pulse generator generates an EM step-pulse into the transmission cable and the waveguide via a multiplexer, and the transmission cable causes the EM wave reflection signal to be displayed on the oscilloscope due to the discontinuity of electrical characteristic impedance (e.g., changes of the geometry or internal material of the cable/waveguide).

Based on the different design of the waveguide, TDR is applied to various monitoring applications, such as a TDR landslide monitoring system with multi-functions as shown in Figure 2 [38]. Generally, TDR can be divided into three types in engineering monitoring applications. The first one is Crimp Type, where the waveguide is the cable itself for landslide monitoring. The sensing waveguide is an off-the-shelf coaxial transmission line and is mainly placed into a borehole with the grout in a soil/rock slope. When the slope slides, it changes the geometry of the coaxial transmission line, causing the reflection spike, indicating the corresponding depth and degree of sliding, as shown in Figure 1. The reflected waveform with spikes is then transmitted and analyzed via wire/wireless communication for landslide early warning. Chung et al. [39] proposed a solution using a time-differential method between reference and measured TDR waveforms as well as a noise-based criterion to automatically identify the level and location of the waveform spike due to the sliding surface.

Although TDR landslide monitoring has high temporal and spatial resolution as well as high measurement sensitivity of detecting the thin shear surface, the landslide dip direction cannot be determined with a single borehole. Nevertheless, the quantification of shear displacement of landslide is case-dependent [18]. Thus Chung et al. [39] discovered that using time variations of the TDR reflected spikes at certain depths can be practical for landslide early warning and evacuation of residents nearby.

The second one is Interface Type, which measures the reflection at the interface between different materials, such as bridge pier scour [3,4,5,6], water pressure and water level [8,9,10]. The third one is Dielectric Type. This type is commonly used to quantify different dielectric properties of the medium, such as the dielectric permittivity for soil moisture [11,12,13,14,15], EC [16,17,18], and SSC [7].

In current operation of the TDR device, such as TDR100/200 (Campbell Scientific) or TDR3000 (Sympuls Aachen), several settings are required to capture the reflected EM waveforms. For example, START POSITION is the beginning of an observation window of a captured waveform for TDR3000, as shown in Figure 1; TIME STEP is the sampling time interval between two samples of a captured waveform; NO. of SAMPLES is the number of data samples of a captured waveform; CHANNEL is the channel of the multiplexer. Furthermore, AVG means the stacking number of TDR waveforms for noise reduction. These parameters are required for TDR manipulation and as references of the subsequent TDR waveform analysis among the aforementioned three measurement types.

## 3. OGC-SWE Initiative

OGC defines the SWE standard that can be applied to different fields with sensors, transducers, and sensor data repositories discoverable, accessible and useable via the Web, and that can be read and controlled remotely. Based on the concept of SWE, the relevant specifications [28] can be divided into information models and schema, and services through the coding descriptions, supporting the Web service agreement and exchanging various service information, as shown in Figure 3. SWE Common defines modules that exchange sensor-related data with each other under the SWE framework [28]. SensorML provides the framework of describing metadata of sensors and systems in an XML schema [30], meanwhile, the O&M defines models of the sensor measurement and observation in an XML schema as well [40]. This paper firstly involves the SensorML to reveal the sensor (waveguide) characteristics and process of data acquisition using TDR.

Web Notification Service (WNS) uses non-synchronous transmission to assist forward notifications of other service sensor information in SWE to the client, and SOS provides a standard to define the service interface for requesting, filtering, and retrieving the sensor system information and sensor-related data observation [31]. Please refer to Section 3.3 for details of SOS.

Sensor Planning Service (SPS) is the service interface for sensor work planning. Users can search for tasks through the SPS service interface, and then the SPS assigns tasks to the sensors. When the sensor task is completed, the SPS uses WNS to notify the user and places the data into the SOS. Finally, Sensor Alert Service (SAS) provides a subscription sensor to generate data, and users subscribe the sensor information through the query condition on SAS. When the sensor generates data and transmits it to the SAS, the SAS refers the alert setting conditions by users and then notifies the user through the WNS transmission.

### 3.1. Sensor Model Language (SensorML)

SensorML, which is one of several implementation standards produced under OGC’s SWE activity as previously mentioned, focuses on the process of measurement and observation, rather than on sensor hardware. Thus, SensorML provides a robust and semantically-tied means of defining processes and processing components associated with the measurement and post-measurement transformation of observations, including the sensors, actuators, as well as computational processes applied pre- and post-measurement. 

The standard of SensorML v2.0 proposes sensor metadata models and an XML implementation. The conformance rules for the XML implementation are based on XML validation using XML Schema representation of SensorML together with processing constraints expressed using Schematron assertions and reports, in which Schematron is well-known as a structural based validation language, and it is as an alternative to existing grammar based approaches [30].

### 3.2. Observation and Measurement Standard (O&M)

O&M standard defines model and related XML schema to describe information of sensor observations and measurements. The O&M standard was published as ISO 19156: 2011 [41]. The standard was originally implemented in XML format, as well as the JSON format standard which was released in 2015 [42]. The standard can be extended be implemented with RESTful transmission method [32], which describes a process from the act of observation to results across a wide variety of application domains. Based on the above advantages, O&M is a core standard in SWE and is a key module for exchanging information. Core class diagrams of O&M are then defined as shown in Figure 4. OM_Observation, as the fundamental class, supports five attributes, six associations, and four conditional restrictions. The main attributes contain:parameter (optional): use NamedValue to describe. This is for arbitrary event-specific parameters, e.g., instrument settings.phenomenonTime (mandatory): use TM_Object to describe. This is the time that the result applies to the feature of interest, or the acquisition duration.resultTime (mandatory): Use TM_Instant to describe the format. This is the completion time of the entire observation program, analysis, and simulation.validTime (optional): use TM_Period to describe. This is the time period during which the result is intended to be used.resultQuality (optional): Use DQ_Element (ISO 19115-1: 2014) [43] to describe the quality of the result, which can be a supplementary description of the procedure.

The associated description contains:metadata: This describes general properties such as the data identifier, the downlink and archiving information to the observation.procedure: This is referenced as the OM_Process class in the procedure. Mainly defines the description of factors affecting the observations, such as platform/instrument/sensor and event operators and algorithms used for the acquisition and the acquisition parameters [29]. OM_Process does not require definition of attribute and is an abstraction category.result: This is the final result of observation.featureOfInterest: Describe real-world measurement objects that is being observed such as rivers, bridges, regions, etc. There is no limit in determining the observed target to its description.observedProperty: Define the properties of the featureOfInterest that are being observed or acquired by the procedure.relatedObservation: This description represents zero or more OM_Observations, where their relationship is essential to provide more context.

The restrictions include:The observedProperty is considered as the property or phenomenon whose value is described or estimated through observation and must be related to the featureOfInterest.The procedure and result must be consistent with the observedProperty.The name of parameter must be unique.

### 3.3. Sensor Observation Service(SOS) 

SOS mainly improves the heterogeneous sensing data format and provides access to observations from sensors and sensor systems by proposing a standard service interface. SOS is based on SOAP (Simple Object Access Protocol) and key/value pair transmission methods, where SOAP is an XML-based messaging protocol specification for exchanging structured information in the implementation of web services [44]. SOS uses standard service interface and standard description format to represent sensor observations and measurements in an interoperable manner [31].

SOS began in 2007 with the release of the SOS 1.0 standard [45], defining three sensor service standards in Core, Enhanced, and Transactional. Core provides core observations. Core contains GetCapabilities, DescribeSensor, and GetObservation. GetCapabilities provides detailed information about this SOS server such as service unit, network interface communication protocol and registration of sensor information. DescribeSensor provides the query function of the registered sensors or sensor systems represented by the SensorML specification. GetObservation provides the service access to sensor observations with the filtering through ranges of time and space domains, Procedure, featureOfInterest, and observedProperty. Transactional imports information such as sensors and observations. Enhanced provides an augmented query method in addition to Core. In the past, the standards and guidelines that refer to modeling of the observation procedure were proposed as SOS v1.0. Song et al. conducted urban environmental monitoring through low-cost sensors (including temperature, humidity, air quality, and brightness) and proposed data open methods and formats [46]. Sorg and Kunkel developed an access standard for aeronautical or raster data in SOS v1.0 and revealed a data module which can provide effective access to metadata and databases [47].

This study follows the SOS v2.0 [31], which is based on four specifications: Core, Transactional, Result Handling, and Enhanced Operations. Result Handling is a new category of SOS 2.0. Result Handling can import by self-developing template to reduce transmission bandwidth. In addition, SOS v2.0 strictly follows OGC Web Service Common Specification [48], SWE Service Model 2.0 [49], Filter Encoding [50] and O&M [41] Pearlman et al. proposed the monitoring data representation format for the Oceans of Tomorrow (OoT) project [51] based on SOS v2.0. Additionally, Sofos et al. defined the land survey data module using SOS v2.0. This model data group realized multi-point monitoring station spatial information query [33].

Currently, SOS implementations are available via 52° North SOS [52], Map Server SOS [53] and Deegree SOS [54]. Among these, limited SOS functions (GetCapabilities, DescribeSensor, and GetObservation) are provided in Map Server SOS and Deegree SOS, while the 52° North SOS supports all SOS features including Transactional and Enhanced Operations. Thus, this study applies the 52° North SOS in the experiments.

## 4. Definition of TDR Model Profile for SOS

Based on the aforementioned TDR monitoring technique (e.g., water level, SSC, and deformation) and the SOS-related methods that have been included in the previous sections, this section explores TDR model profile for SOS. First, the fundamental framework the TDR model profile is composed of is as follows:Procedure, Offering, and observableProperty by providing the registration methods of TDR Project info, Station, and Measured Property;Class diagram of TDR OM_Process conceptual model;Insert sensor with XML request.

All these points follow the definition of O&M in the feature of interest-observed property-process-result. Following sections will discuss the allocation method and finally implementation of the architecture of the data module through the SOS v2.0.

### 4.1. Registration of Procedure, Offering and obeservableProperty

Due to the consideration of the required information of the project which consists of the monitoring stations and subsidiary TDR monitoring sensor types, this study establishes a TDR registration method in SOS. Figure 5 first depicts the definition of the SOS, consisting of Project Info, Station, and Measured Property. One or more monitoring stations may be included in a proposed project, and one or more TDR sensor types, such as SSC, water level, and landslide monitoring, would be included in a single monitoring station. The measurement by each TDR sensor type in a monitoring station is unique, thus Project Info, Station and Measured Property are referenced as the SOS identification codes of Procedure, Offering and observableProperty, respectively.

### 4.2. Class Diagram of TDR OM_Process Conceptual Model

This study uses OM_Process to define basic information about TDR monitoring processes, such as SSC (TDR_SSC_Process), water level (TDR_WL_Process) and conductivity (TDR_EC_Process), as shown in Figure 6. OM_ComplexObservation is further applied to the crimp type (TDR_Deformation_Process) due to the acquired waveform feature of TDR landslide application. These classes inherit the OM_Process class, and each monitoring process includes monitoring device (OnsiteDevice), operator information (Operator), station information (StationInfo), and TDR device setting information (TDRSetting) as previously mentioned. The following defines the parameters in detail:Field device information (TDR_Device): device name (Name) and device model (Model), such as TDR3000 (Sympuls Aachen) or TDR100/200 (Campbell Scientific).Operator Information (TDR_Operator): Name, Email, and the phone number of Contact.Station information (TDR_Deformation_Info, TDR_SSC_Info, TDR_WL_Info, TDR_EC_Info): According to different TDR monitoring methods, different station information is proposed. First, the crimp type monitoring (TDR_Deformation_Info), as referred to in Figure 6, includes station name (Title), waveform filtering method (FilterType), depth of slope (ElevationOfStart), waveform effective range (StartPoint and EndPoint), depth conversion coefficient (DepthInterval), reference waveform (RefWave) and sliding threshold value (Threshold). Second, suspended sediment concentration monitoring (TDR_SSC_Info) contains the station name (Title), warning threshold (Threshold) and probe depth (Depth). Third, water level monitoring (TDR_WL_Info) contains the station name (Title), the water level elevation (Elevation) and the warning threshold (Threshold). Final, EC monitoring (TDR_EC_Info) contains the station name (Title) and the warning threshold (Threshold).TDR host setting information (TDR_Device_Setting): TDR mainframe is divided into two models, TDR3000 (Sympuls Aachen) or TDR100/200 (Campbell Scientific). TDR100/200 setting includes captured window range (Start and Length), total number of waveform data (NumOfData), and average number of waveforms for stacking (Avg). TDR3000 setting includes start point of waveform (Start), time interval (TimeInterval), total number of waveform data (NumOfData), and average number of waveforms for stacking (Avg).

Figure 7 shows the classes of TDR observations inherited from OM_ComplexObservation, which include landslide deformation observations (TDR_Deformation_Observation), suspended sediment concentration observations (TDR_SSC_Observation), electrical conductivity observations (TDR_EC_Observation), and water level observations (TDR_WL_Observation). The following describes the definitions of TDR observations: TDR_Deformation_Observation: This class includes TDR measurement waveforms (Wave) in SWE DataArray format and sliding alarm data. Information of sliding warning includes sliding scale (AlarmLevel) and corresponding sliding depth (AlarmDepth).TDR_SSC_Observation: This class contains the temperature of water (Temperature), the standard deviation of the temperature in water (TemperatureSTD), SSC (Concentration) and the standard deviation of the SSC (ConcentrationSTD).TDR_EC_Observation: This class contains the standard deviation (ECSTD) of EC and EC itself (EC).TDR_WL_Observation: This class includes water level (WaterLevel) and water level standard deviation (WaterLevelSTD).

### 4.3. Insert Sensor

Figure 8 demonstrates the beginning of an InsertSensor request. Lines 1–5 indicate the specification version and namespaces of each referenced specification. Lines 6–7 use PhysicalSystem in SensorML v2.0 to indicate the sensor information. The identifier is to uniquely identify the sensor and this study defines the identifier as the released project name as shown in Figure 5. Lines 8–23 include the full name and abbreviation of project, where names of the research unit and project can be placed. In lines 24–33, the Offering is an important discriminating identifier for managing the stations of the project. This study defines an offeringID as the name of the monitoring station. Lines 34–39 represent the featuresOfInterest, which is a description of the observed feature. Observed features can be bridges, buildings, rivers, or air as mentioned previously. There is no restriction on this featuresOfInterest description in the SensorML 2.0 specification. This study uses the name of the monitoring station as the identifier of the observed feature, where geographic location of each monitoring station should be reported in the InsertObservation request.

Figure 9 shows the output properties. These help describe measurement parameters (ObservableProperty) of various TDR measurement types with the identifier of the monitoring station name. Each outputList contains multiple output tags. The output tag contains the name of the measurement parameter (name) and the corresponding URI definition (definition). Users can follow the URI to learn more about the TDR measurement parameters.

Regarding the format of operator information, SensorML 2.0 specification states that Contacts attribute, in the history events category, can be used to describe the manufacturer, domain expert, or anyone involved. Since the 52° North SOS implementation does not support this format, this study refers to and defines operator information in the PhysicalSystem with the parameter attribute. As shown in Figure 10, the operator information is described with the SWE DataArray format. Lines 1–10 describe the DataArray content, including basic description (swe: label and swe: description) and number of data in each record (swe: Count). Lines 11–39 define each parameter, where the definition URLs provide relevant descriptions. Lines 40–47 include the values corresponding to the defined parameters in the DataArray format.

The Physical Component category in the SensorML 2.0 specification can be used to describe any device, including overall process through the Input and Output attribute tags. However, the 52° North SOS implementation does not support this format either. This study proposes a solution similar to the idea of the operator information. A general format is shown in Figure 11.

To be specific, the TDR landslide monitoring information parameters, as shown in Figure 12, include the Name of the station, FilterType (waveform filtering method), ElevationOfStart (starting depth), StartPoint (the starting data point of the effective range of the measuring waveform, unit: positive integer), EndPoint (measure the end of the effective range of the waveform, unit: positive integer), DepthInterval (the coefficient of the depth of the waveform point conversion, positive integer unit), RefWave (reference waveform) and Threshold (the threshold value of the slope sliding, unit: reflection coefficient ranging from –1 to 1). In addition, the TDR SSC monitoring information parameters are shown in Figure 13, including Name of the station, Depth (monitoring total water depth, unit: meter) and Threshold (the threshold for mud concentration exceeding the standard, unit: parts per million).

Furthermore, the geographic location of the monitoring station is shown in Figure 14. The lines 1–8 and 9–16 indicate the longitude and the latitude of the sensor in plan view, respectively. For the case such as a directionally drilled TDR cable, the coordinates of the control points of the cable are suggested to be provided sequentially to reveal the line feature in space. Finally, the ObservableProperty and metadata description are shown in Figure 15. Lines 1–12 include the ObservableProperty measured by the TDR station. Since the SOS specification states that ObservableProperty content must provide a URI definition, this study hosts the URLs in Figure 15 to describe each ObservableProperty content. Lines 13–24 represent the format specification for InsertObservation supported by this SOS version.

### 4.4. Insert Observation

As mentioned in the previous section, a registered sensor can use the InsertObservation operation to publish measured observations on an SOS service. The InsertObservation request requires four identifiers registered to the SOS via the InsertSensor operation, including Procedure, Offering, FeatureOfInterest, and ObservableProperty. First, Procedure is content of the Identifier tag of the InsertSensor request. Offering is the monitoring station, FeatureOfInterest is measured target and ObservableProperty defines the monitoring types (e.g., measured temperature for TDR SSC monitoring). Thus, this study defines InsertObservation using ComplexObservation method and presents an InsertObservation request of TDR landslide monitoring observations (Figure 16 and Figure 17) as an example.

Lines 1–16 in Figure 16 reveal the format based on the aforementioned specification, as well as the namespace for each referenced specification. Line 17 provides the offeringID that has been registered to the SOS via the InsertSensor. Lines 21–26 represent the phenomenonTime, which is the time of the measurement. As the resultTime is the time that an observation result was produced, TDR creates results at the same time that an observation was measured. Hence, the resultTime is the same as the phenomenonTime. Since TDR observations happen at individual time points, this study uses TimeInstant to describe the measurement time.

Line 27 indicates the Procedure of this observation, which was measured to the SOS via the InsertSensor. Line 28 shows ObservedProperty of this observation, which should also follow a registered ObservedProperty in the InsertSensor. Lines 29–41 represent the FeatureOfInterest of this observation, where the name tag of identifier indicates the station name, and SampledFeature describes this feature. For example, we define a project name as Test, station name as TestStation, and measurement type as Deformation. The actual geographic location is revealed in the Geography Markup Language (GML) geometry format. GML is the XML grammar defined by the OGC to express geographical features [55]. Line 42 is the actual measurement data which uses string data type to represent a TDR deformation wave observation. The data content is divided into reflected signals by using a comma (,), and arranged in order according to the measured depth. If users want to know the depth of measurement corresponding to each reflection coefficient, they can obtain the measured depths and the conversion coefficient from the DeformationInfo in InsertSensor. For example, the depth of the second point is 2 * 0.025 = 0.05 meters where depth is (Data Point)* conversion coefficient (DepthInterval).

Figure 17 shows the continued content of this TDR displacement measurement including the alert level (Alarm_Level) and alert depth (Alarm_Depth). Each sos: Observation represents an individual observation, where the om:ObservedProperty tag defines the observed phenomenon. In addition, as the phenomenonTime, resultTime, and FeatureOfInterest have been defined previously in the TDR observation, these attributes can simply reference to the defined values. Finally, the data type of result tags is gml:MeasureType, where the uom is the unit of measurement for the result value.

## 5. System Implementation and TDR Data Visualization

With the proposed TDR model, the development connection scheme between TDR monitoring information and an SOS service, as depicted in Figure 18, mainly includes:

1. SOS Management Interface:

SOS Management Interface is a client-side interface for web publishing and data sharing. First, the user can register sensor information through the interface settings of the 52° North SOS, and the interface stores the successful registration settings in the published configuration database.

2. SOS Middleware:

The middleware continuously publishes TDR observations to the 52° North SOS. This intermediary software will automatically query various TDR observations from TDR platform database [24] and finally publish observations to the 52° North SOS service.

3. 52° North SOS service:

This study applies the 52° North SOS implementation to support the two versions of SOS standards [31,52]. To be specific, this study mainly inserts and queries sensor descriptions and sensor observations from an SOS service.

Figure 18 depicts that the user firstly establishes the data sharing settings through the SOS Management Interface and sends registration of TDR monitoring descriptions to the 52° North SOS service through the background procedure of the SOS Management Interface. The SOS Management Interface will access the published configuration database when successfully receiving the registration message from the SOS. The SOS Middleware periodically queries the registration message and observations of the TDR sensor from the TDR database and sends observations to the SOS service via InsertObservation. Consequently, the user can retrieve all registered procedures with the GetCapabilities operation and can also retrieve detailed sensor information via Describe Sensor and obtain sensor observations by Get Observation.

Figure 19 demonstrates TDR web interface to set aforementioned effective range and reference waveform for TDR crimp type monitoring. The date of each monitoring data, which is used to set reference waveform for the time-differential method, is displayed on left side of the page. Then a user can adjust the threshold range to the TDR waveform by multiplying the standard deviation which is obtained from the differential result between measured waveform and the reference waveform. When the cursor moves to the waveform plot, the point number and the corresponding reflection coefficient are displayed simultaneously.

Figure 20 reveals the historical data and sliding trend of TDR landslide monitoring. The part ‘a’ is a raw waveform data. The part ‘b’ demonstrates the results by the time-differential method subjected to the reference waveform, and the red line represents the criteria by setting three times standard deviation ranges of waveform noise level. The part ‘c’ is a sliding spike tendency according to the sliding depth. Overall, based on the proposed TDR model using SOS, all data of TDR landslide demonstrations are stored and hosted in an interoperable manner.

## 6. Conclusions

The Time-domain reflectometry monitoring system has provided a complete and diverse solution from field-side device monitoring to server-side web services. This study attempts to improve the interoperability of TDR sensor metadata and observations by proposing a TDR data model profile based on the OGC Observations and Measurements international open standard. By packaging TDR data with the well-defined O&M conceptual model, TDR data can be represented and understood unambiguously. In addition, this study implements a system prototype transforming TDR data and continuously uploading observations to an OGC Sensor Observation Service to help users retrieve TDR data in an interoperable manner. Overall, we believe that by standardizing and sharing TDR data based on the OGC Sensor Web Enablement standards, TDR data can be effectively and efficiently utilized by many important applications, such as landslide monitoring and related management based on this TDR landslide early warning system.

## Figures and Tables

**Figure 1 sensors-19-04070-f001:**
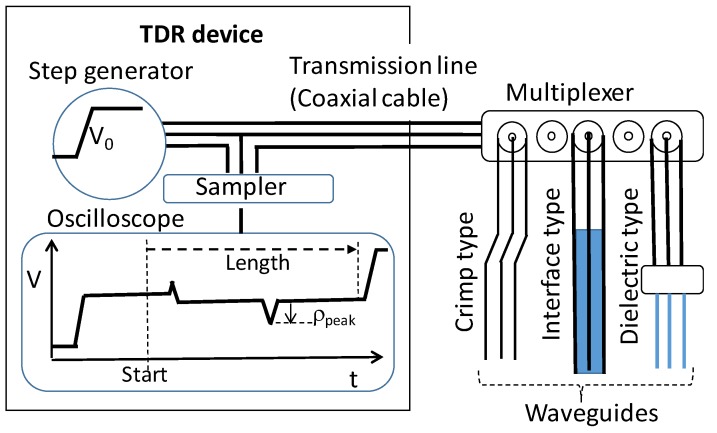
Time-domain reflectometry (TDR) schematic and related sensing waveguides.

**Figure 2 sensors-19-04070-f002:**
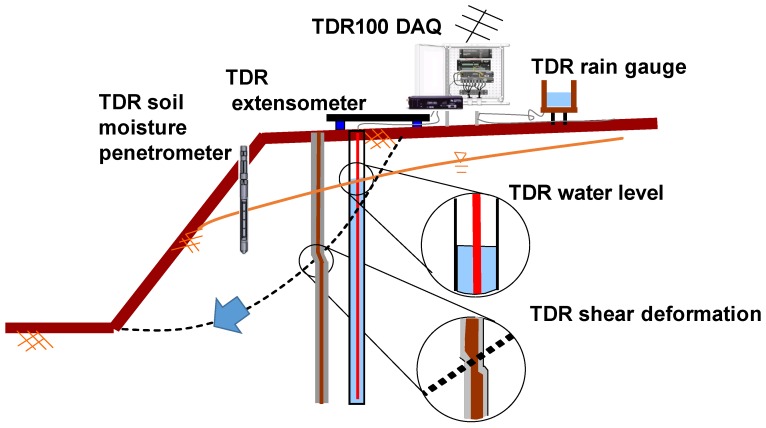
TDR landslide monitoring system with multi-functions, adapted from [22].

**Figure 3 sensors-19-04070-f003:**
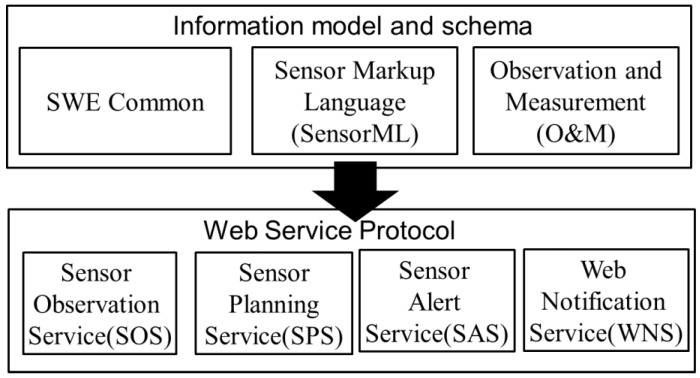
Framework of Sensor Web Enablement (SWE).

**Figure 4 sensors-19-04070-f004:**
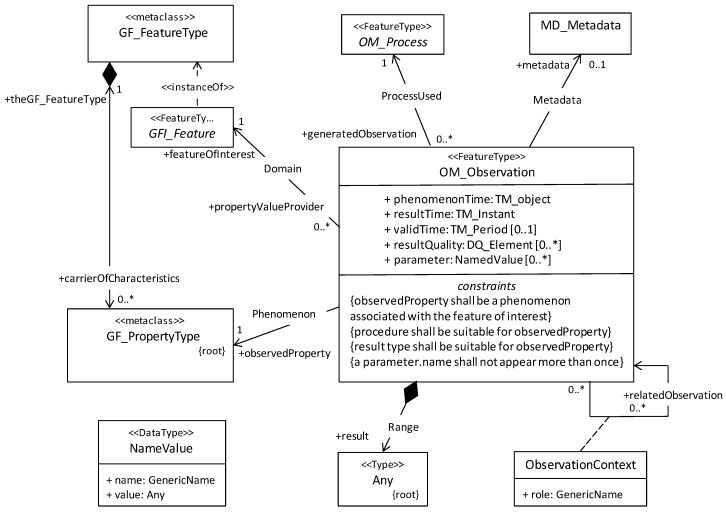
Core class diagram of Observation and Measurement O&M conceptual model [29].

**Figure 5 sensors-19-04070-f005:**
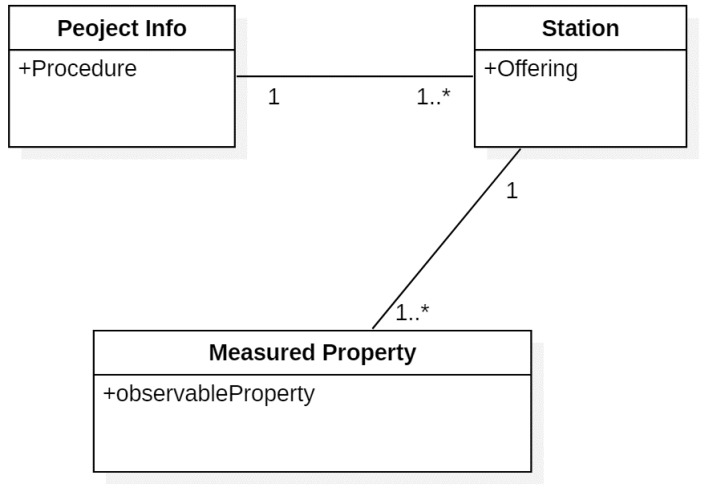
Definition of the relationship diagram of the SOS registration code for TDR.

**Figure 6 sensors-19-04070-f006:**
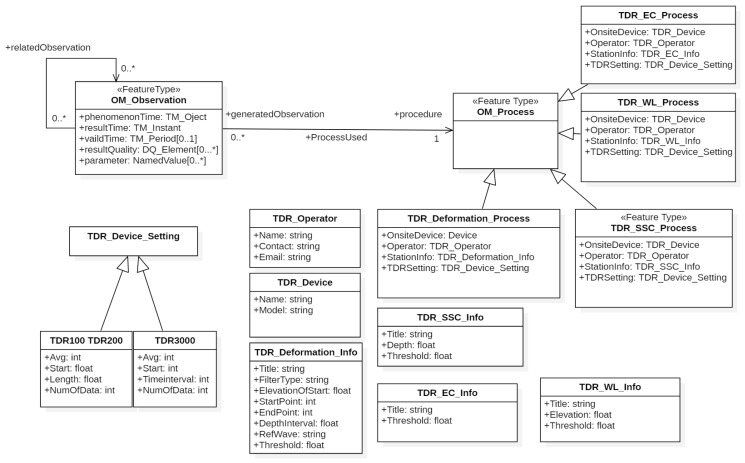
Class diagram of TDR OM_Process conceptual model.

**Figure 7 sensors-19-04070-f007:**
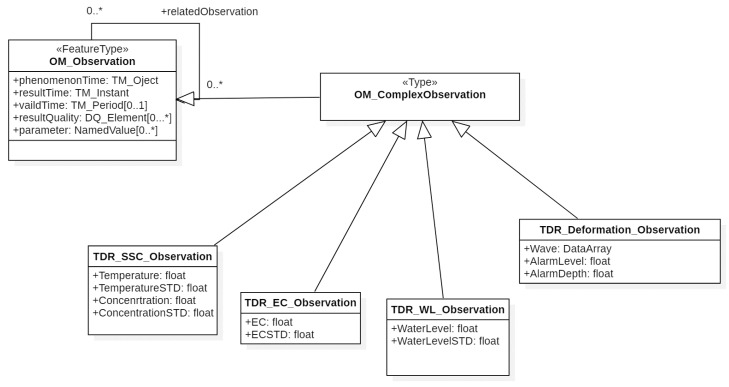
Class diagram of TDR OM_ComplexObservation conceptual model.

**Figure 8 sensors-19-04070-f008:**
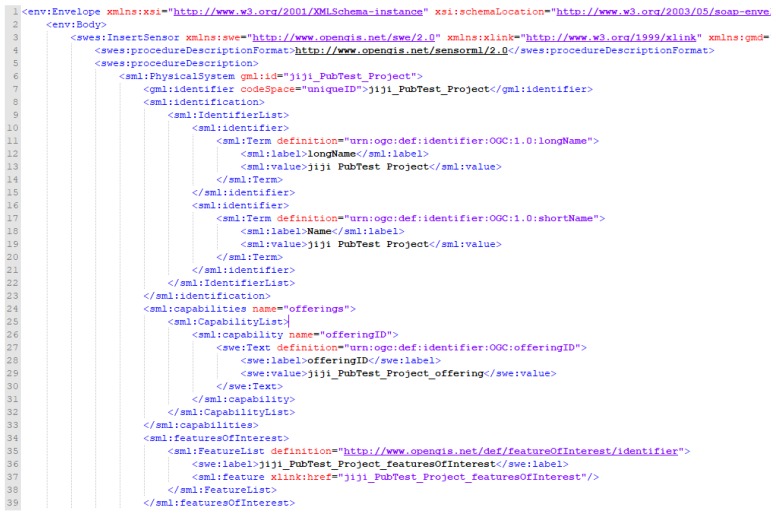
Insert sensor Extensible Markup Language (XML) request, InsertSensor_identifier, offering, and featuresOfInterest.

**Figure 9 sensors-19-04070-f009:**
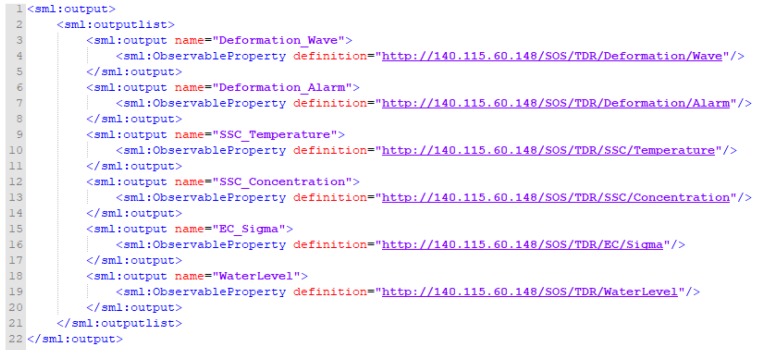
Insert sensor XML request of InsertSensor_outputs.

**Figure 10 sensors-19-04070-f010:**
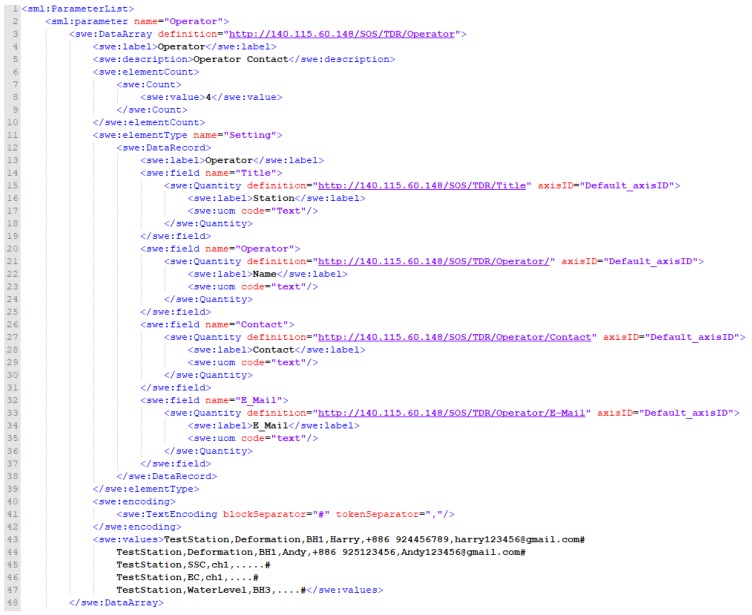
Extended InsertSensor format for Operator.

**Figure 11 sensors-19-04070-f011:**
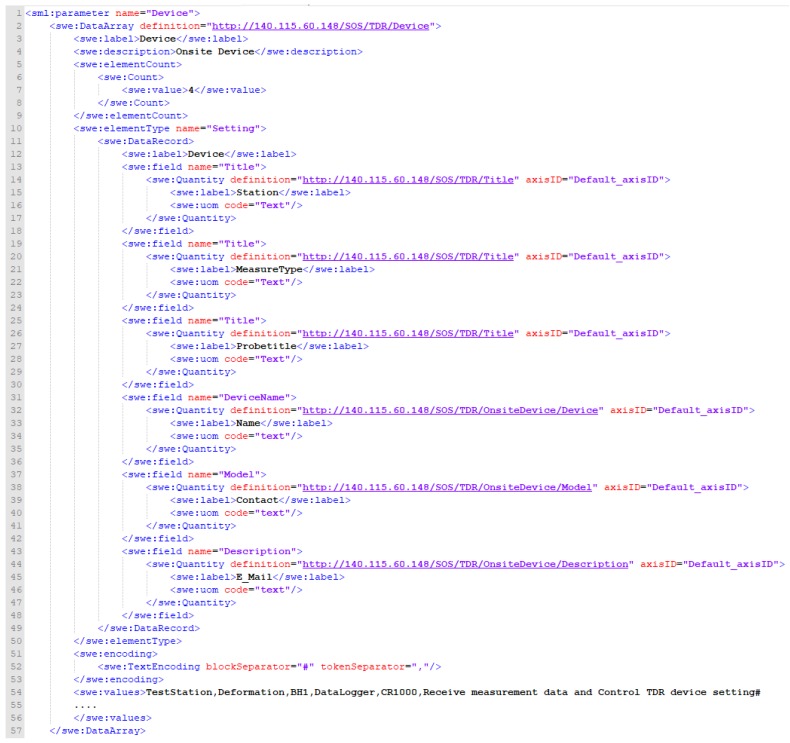
Extended InsertSensor format for InsertSensor_Device

**Figure 12 sensors-19-04070-f012:**
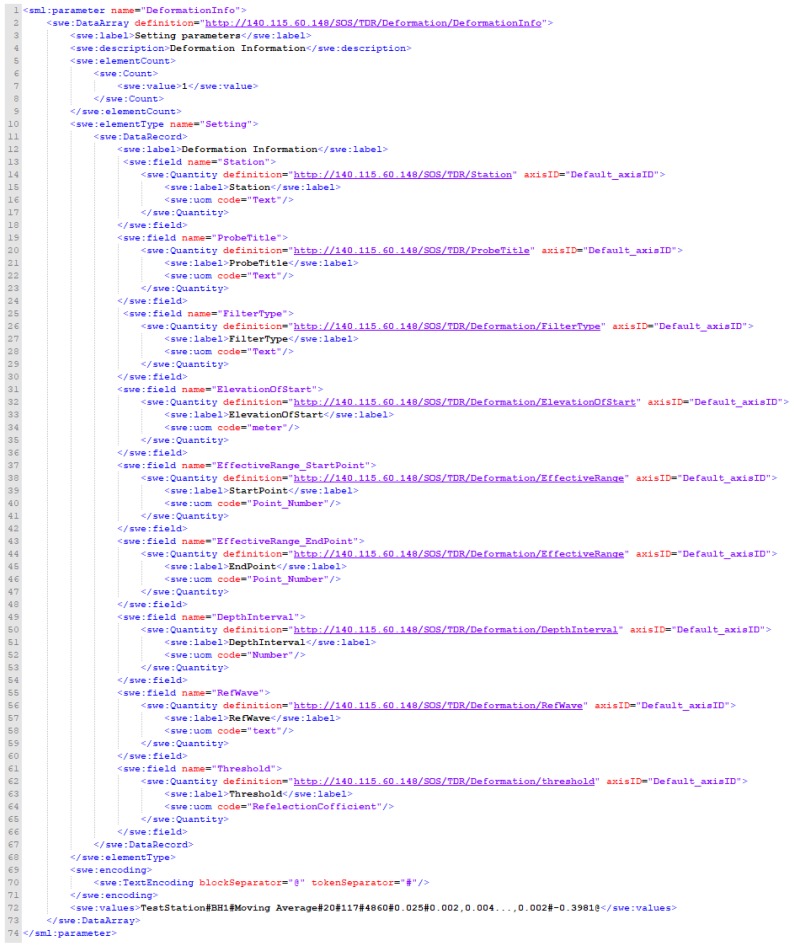
Insert sensor XML request of InsertSensor_DeformationInfo.

**Figure 13 sensors-19-04070-f013:**
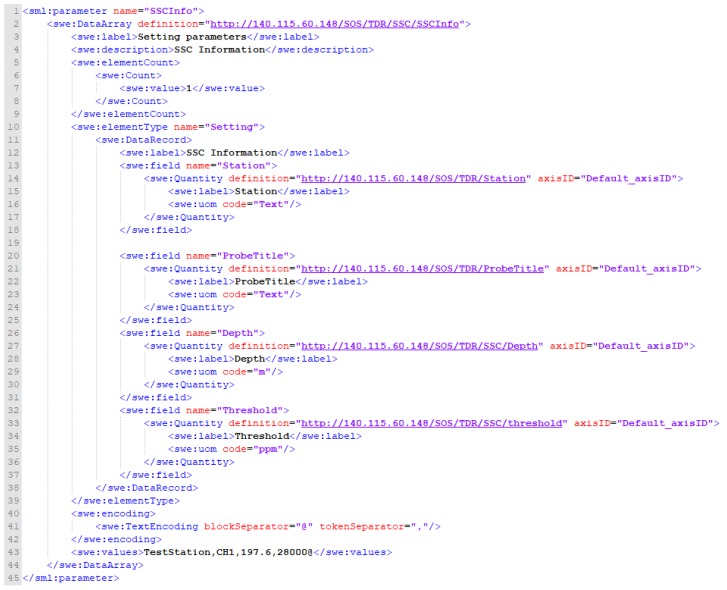
Insert sensor XML request of InsertSensor_SSCInfo.

**Figure 14 sensors-19-04070-f014:**
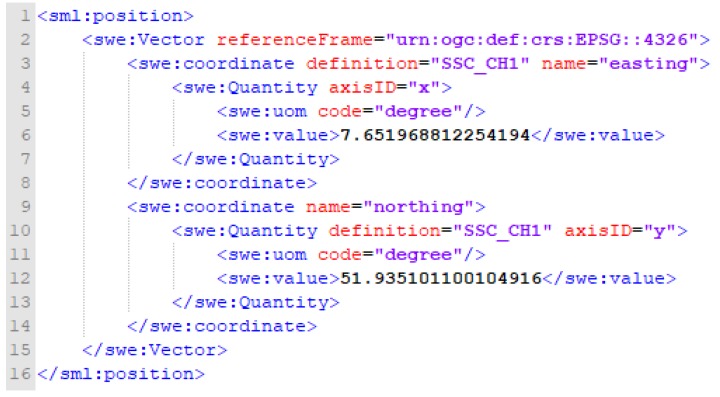
Insert sensor XML request of InsertSensor_position.

**Figure 15 sensors-19-04070-f015:**
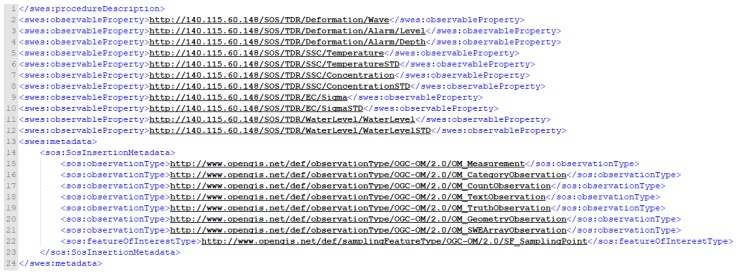
Insert sensor XML request of InsertSensor_observableProperty&metadata.

**Figure 16 sensors-19-04070-f016:**
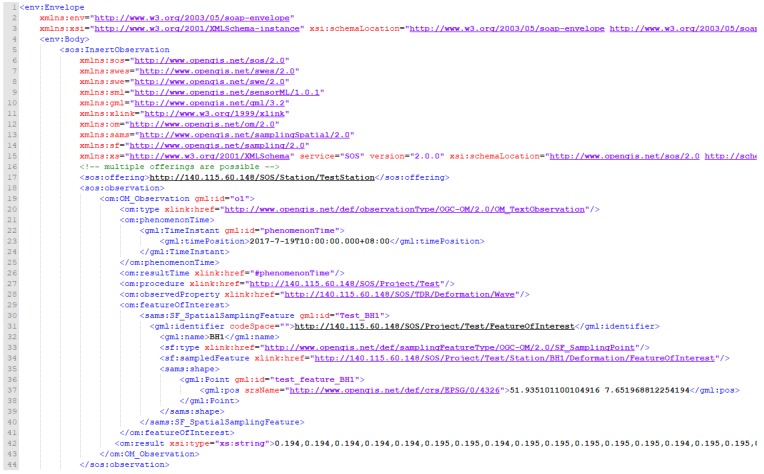
Insert Observation XML request of TDR_Deformation_Observation (Part I).

**Figure 17 sensors-19-04070-f017:**
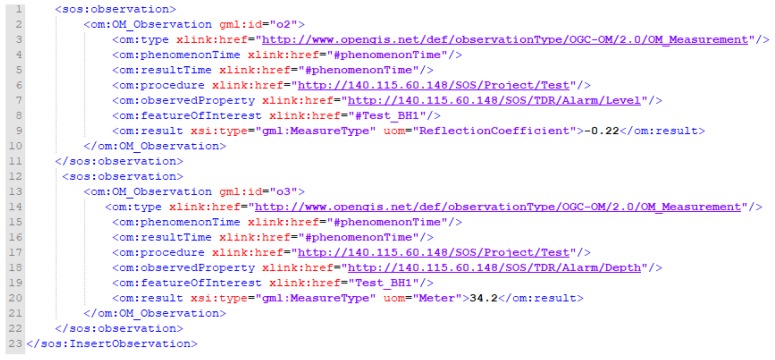
Insert Observation XML request of TDR_Deformation_Observation (Part II).

**Figure 18 sensors-19-04070-f018:**
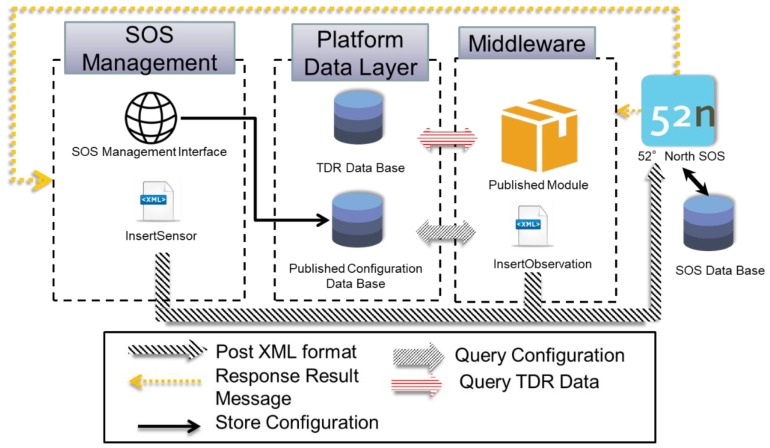
Connection schema between TDR information system and SOS.

**Figure 19 sensors-19-04070-f019:**
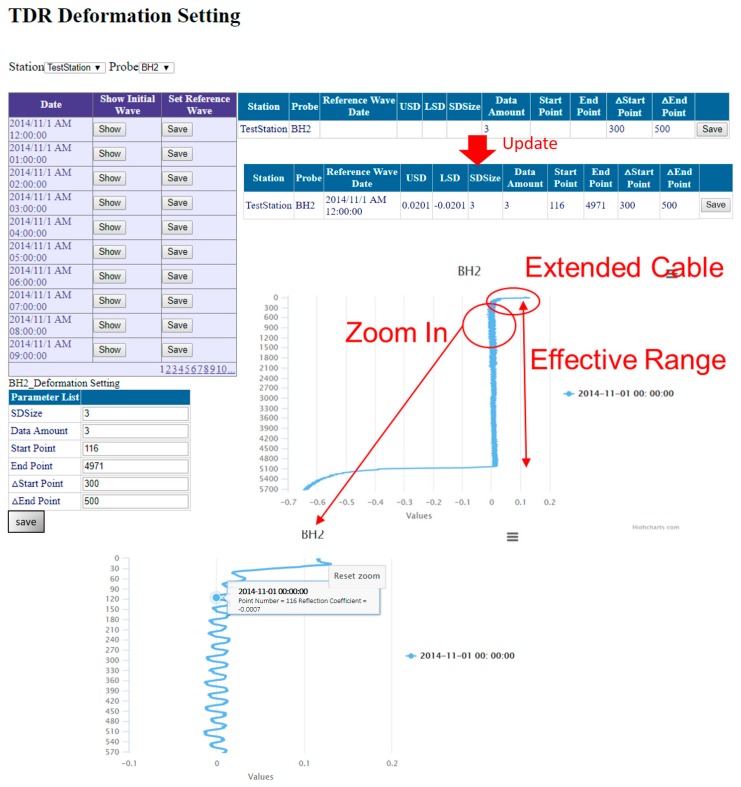
TDR crimp type monitoring setting for reference waveform and effective range.

**Figure 20 sensors-19-04070-f020:**
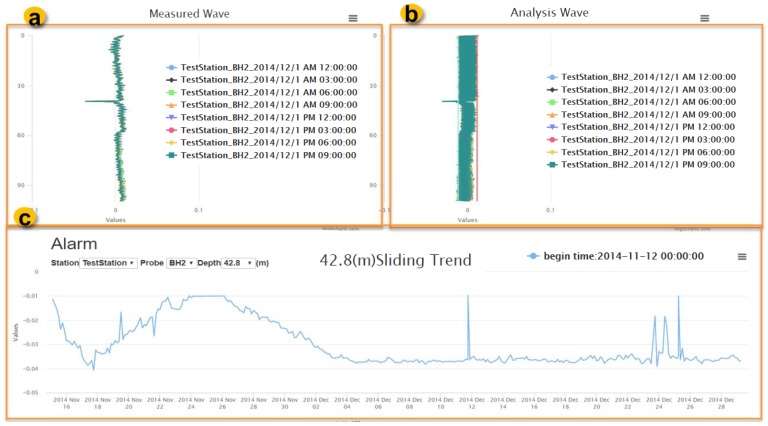
(**a**) TDR shear deformation historical raw waveform data, (**b**) waveform differential results, and (**c**) sliding trend at 42.8 m depth.

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
