# Peer review of "Applying OGC Sensor Web Enablement Standards to Develop a TDR Multi-Functional Measurement Model"

_sensors, 2019, doi:10.3390/s19194070_

Round 1

Reviewer 1 Report

Review of sensors-583215 Applying OGC Sensor Web Enablement Standards to Develop a TDR Multi-Functional Measurement Model

The authors present an information model for inclusion of time-domain reflectometry (TDR) data from geotechnical engineering and related applications in a generalized measurement framework. The proposed model extends the existing classes and XML models promulgated by the Open Geospatial Consortium. The authors include sample data requests and visualizations supported by the proposed model.

I found the proposed model to be intriguing and promising. The manuscript is generally well organized and written.  The figures are generally clear and each adds something to the document. The authors present a compelling case for data inter-operability in the geotechnical instrumentation domain and present a proposed solution that builds on existing work.

OVERALL COMMENTS

Beginning in the abstract, the authors mention “disaster prevention” and “disaster management”. Prevention or mitigation of disasters is a noble and appropriate goal – after all, as engineers, our first responsibility is to the public good – but I think the focus and flow of the abstract would be considerable improved by leading with a description of the authors’ original work. Follow up with the motivation later in the abstract.

Code listings (e.g. Figures 10-14): The code listings were helpful; however, I would encourage the authors (or editors, as appropriate) to present the code listings as text in a dedicated listing environment (with line numbers) rather than as inline images in Figure environments. In particular, the code listing in Figure 12 is blurry and difficult to read due to its inclusion as an inline image.

The paper is well organized and the written English is of generally good quality. However, some minor grammatical errors (e.g. subject-verb or subject-adjective agreement) are pervasive. There are also several instances of disagreement in spelling of class names between the UML diagrams and the narrative (e.g. Figure 6).  Please consider engaging a third-party proofreader to eliminate these mistakes. (I find that these sorts of errors creep up in my own work as well – familiarity with the manuscript seems to make these issues “invisible” to the author).

Provided it is acceptable to the editor/copy editor, I recommend typesetting class and function names in a monospaced font (e.g. Courier) rather than write them within single quotes.

Aside from a passing reference to the 1999 Dowding-O’Connor book, the literature review in the Introduction section appears to refer almost exclusively to the authors’ own work in the past ten years or so. It is important to demonstrate how the work described in the manuscript fits within the broader literature, not just the authors’ recent work.  What’s more, additional references to case studies and specific challenges encountered in geotechnical / soil science applications of TDR would demonstrate what kinds of engineering decisions have been / may be supported by TDR measurements and thus strengthen the authors’ arguments for effective data interchange.  Dowding’s web site is a good place to start a review of literature from the 1990s and early 2000s. Many of the articles are available as PDFs on his site (http://iti.northwestern.edu/tdr/pubs.html). Among those articles are seminal pieces on application to slope stability, bridge pier scour, and ground water.  O’Connor has also published a number of case studies on highway and bridge applications, as well as mine subsidence issues. Many geotechnical applications by Kane and Beck were described in the mid-1990s.  Use of TDR for soil moisture measurement dates back at least as far as 1980. To provide a full picture of where current work is positioned in the literature, it is useful to make at least some brief references to seminal work in the field as well as recent work.  Other places to look for TDR case studies include the proceedings of the TDR conferences in 1994 ((Foulum, Denmark), 2001 (Evanston, Illinois, USA), and 2006 (West Lafayette, Indiana, USA).

LINE-BY-LINE COMMENTS

Line 26: “Internet” is a proper noun and should be capitalized.

Line 38: It is worth noting that data transmission of TDR and related instrument data from a bridge to an Internet-accessible database for near-real time interpretation dates back at least as early as 1999. I recall work by Marron, Dowding, O’Connor, and others on bridges in California and Indiana, USA, during that era. Dowding’s lab put considerable effort into developing some early data interchange models for TDR and related instrument data in the mid-2000s; some of this work was summarized in thesis by Dussud in 2001 and by articles by Kosnik et al in the proceedings of Field Measurements in Geomechanics 2007 (ASCE Geotechnical Special Publication 175). Thus, while the authors have correctly stated that TDR analysis is complex and that well-considered models for interoperability are useful and necessary, it is not accurate to say that no interoperable models existed before this current work. (I do agree that the authors’ work is novel and innovative – it’s just not the first effort in this area).

Line 44: “Hub” is capitalized as a proper noun. Is this a trade name, standard, or similar? If so, please define and provide a reference.

Line 47: Referring to something (open standards, in this case) as the “ultimate” approach smacks of hyperbole. Please use a less loaded term, such as “attractive” or “useful”.

Line 50: Similarly, unless it can be quantitatively proven, please avoid phrases such as “the most comprehensive”. “..as a comprehensive solution” would be sufficient.

Line 64: I recommend moving the introduction of XML to about Line 52 (since SensorML and WaterML are XML derivatives, it is useful to discuss XML in general before mentioning SensorML, ec)

Line 67: Please define “REST” here – not all readers will be familiar with how HTTP and architectures built upon it work.  It is mentioned again on Line 71 few lines later, and again much later in the document in terms of a RESTful API – please provide a very brief definition and link to a reference document (I think it was defined in a dissertation circa 2000). It would probably be sufficient to say that REST provides a means for stateless information exchange over HTTP for Web services.

Line 71: Provide a reference to the JSON standard or reference implementation if possible

Line 74:  “It is believed….” Believed by whom?  “…that XML represents the most basic”  Again, “the most” should be omitted unless it can be demonstrated quantitatively.

Line 85: Consider mentioning a popular concise definition of TDR:  “cable radar” or “radar along a cable”

Line 96: Consider replacing “only a common” with “an off-the-shelf” coaxial transmission line. Strike “and mainly” and replace with “commonly”.

Line 105: it would be more accurate to refer to the failure surface as a “shear surface” rather than “fracture layer”

Line 115: The manufacturer of the TDR100/200 units is listed; can you list a manufacturer for the TDR3000/SYMPULS?

Line 164: please define “Schematron”

Line 169: “Restful” should be written “RESTful” to refer to the HTTP architecture.

Line 195: “There is no limit to its description”. No limit in what sense? Size of the data structure? Please clarify.

Line 198: Consider phrasing “represents none to multiple” as “represents zero or more” to align with common usage in UML cardinality descriptions

Line 201: What is meant by “considered as a phenomenon”?

Line 204: Consider phrasing “cannot be repeated” as “must be unique”. In general, it is better to state what something IS rather than what it IS NOT.

Figure 4: This figure is difficult to read when printed in grayscale. Please consider removing the color backgrounds from the UML class rectangles. At a minimum, use a different color scheme than red text on a pink background for the parameters of the central OM_Observation class.

Line 212: Please provide a brief (one sentence) description of the SOAP protocol, and link to a specification or reference document. Strike “KVP”, as this abbreviation is never used again.

Line 239:  Is the SOS “Degree” or “Deegree”? Spelling is inconsistent between the narrative and the reference list.  Also, the sentence seems to suggest that “based on Java” and “open sourced” are mutually exclusive, which they are not. Please clarify. Do you mean to suggest that the former is proprietary?  If so, the language choice is irrelevant.

Line 245:  “water level” and “deformation” are not proper nouns and thus should be all lowercase.

Line 261: Is “Measure Property” a noun or a verb (property or method)?

Line 264: Strike “and cannot be repeated”. It is sufficient to say “unique”.

Figure 6 and elsewhere: please check for consistency of spelling and orthography for class/property/method names between the UML diagrams and the narrative text (e.g. TDR_WL_Observation in the diagram vs TDR_WL_Obsrvation in the narrative).

Figure 9, Section 4.3., and related sections:  “Part A”, “Part B”, and so on should be capitalized. Also, consider adding line numbers to code listings and refer to the individual parts by line number.

Line 349: The figure cross reference here is broken.

Line  358:  As the abbreviation “PPM” is never used again, it is sufficient to write “parts per million” (lowercase)

Line 360: How are features that are not points in plan view (e.g. a directionally drilled TDR cable) represented by latitude/longitude?

Figure 8 / Line 366: why is this line highlighted?

Figure 9: why is a line in Part B highlighted?

Figure 11: why is a line near the bottom highlighted?

Figure 12 is blurry and is too small to be legible.

Line 400: “…of this observation, which should have been registered….”   Is there an exceptional case where the Procedure would not have been registered?  If not, it is sufficient to write “…this observation, which was measured….”

Line 406:  This is the first mention of GML. Please define it and provide a citation.

Line 429:  “52o North SOS” – shouldn’t that be a degree sign, rather than a lowercase ‘o’?

Line 430 (and again on 443):  “Published Configuration Database” is capitalized as though it were a proper noun. Is this a particular information model object? If so, where was it defined?  Please clarify.

Line 452: “adjust the threshold range by multiplying the standard deviation to the TDR waveform”. The standard deviation of what? A “noise” signal? Please clarify.

Line 475:  Consider replacing the word “accurately” with “unambiguously” or similar to avoid semantic confusion with “accuracy” in the metrology sense.

Author Response

Responses of Reviewer 1 comments:

The authors present an information model for inclusion of time-domain reflectometry (TDR) data from geotechnical engineering and related applications in a generalized measurement framework. The proposed model extends the existing classes and XML models promulgated by the Open Geospatial Consortium. The authors include sample data requests and visualizations supported by the proposed model.

I found the proposed model to be intriguing and promising. The manuscript is generally well organized and written.  The figures are generally clear and each adds something to the document. The authors present a compelling case for data inter-operability in the geotechnical instrumentation domain and present a proposed solution that builds on existing work.

R: Authors appreciate the comments. Following please find the responses to each comments in detail.

OVERALL COMMENTS

Beginning in the abstract, the authors mention “disaster prevention” and “disaster management”. Prevention or mitigation of disasters is a noble and appropriate goal – after all, as engineers, our first responsibility is to the public good – but I think the focus and flow of the abstract would be considerable improved by leading with a description of the authors’ original work. Follow up with the motivation later in the abstract.

R: Authors appreciate the comments, and the abstract is modified as suggested.

Code listings (e.g. Figures 10-14): The code listings were helpful; however, I would encourage the authors (or editors, as appropriate) to present the code listings as text in a dedicated listing environment (with line numbers) rather than as inline images in Figure environments. In particular, the code listing in Figure 12 is blurry and difficult to read due to its inclusion as an inline image.

R: Authors appreciate the comments. For the consistency, Figure 8~17 were presented with the code listings as text in a dedicated listing environment (with line numbers) with higher resolution.

The paper is well organized and the written English is of generally good quality. However, some minor grammatical errors (e.g. subject-verb or subject-adjective agreement) are pervasive. There are also several instances of disagreement in spelling of class names between the UML diagrams and the narrative (e.g. Figure 6). Please consider engaging a third-party proofreader to eliminate these mistakes. (I find that these sorts of errors creep up in my own work as well – familiarity with the manuscript seems to make these issues “invisible” to the author).

R: Authors appreciate the comments. The instances of disagreement in spelling of class names between the UML diagrams and the narrative in Figure 6 were corrected as suggested. The context has been consulted with the English edit to eliminate these mistakes as suggested.

Provided it is acceptable to the editor/copy editor, I recommend typesetting class and function names in a monospaced font (e.g. Courier) rather than write them within single quotes.

R: Authors modified the content by typesetting class and function names in a Courier font as suggested.

Aside from a passing reference to the 1999 Dowding-O’Connor book, the literature review in the Introduction section appears to refer almost exclusively to the authors’ own work in the past ten years or so. It is important to demonstrate how the work described in the manuscript fits within the broader literature, not just the authors’ recent work. What’s more, additional references to case studies and specific challenges encountered in geotechnical / soil science applications of TDR would demonstrate what kinds of engineering decisions have been / may be supported by TDR measurements and thus strengthen the authors’ arguments for effective data interchange.  Dowding’s web site is a good place to start a review of literature from the 1990s and early 2000s. Many of the articles are available as PDFs on his site (http://iti.northwestern.edu/tdr/pubs.html). Among those articles are seminal pieces on application to slope stability, bridge pier scour, and ground water.  O’Connor has also published a number of case studies on highway and bridge applications, as well as mine subsidence issues. Many geotechnical applications by Kane and Beck were described in the mid-1990s.  Use of TDR for soil moisture measurement dates back at least as far as 1980. To provide a full picture of where current work is positioned in the literature, it is useful to make at least some brief references to seminal work in the field as well as recent work.  Other places to look for TDR case studies include the proceedings of the TDR conferences in 1994 ((Foulum, Denmark), 2001 (Evanston, Illinois, USA), and 2006 (West Lafayette, Indiana, USA).

R: Authors appreciate the comments and we have sited relevant references to support TDR measurements and thus strengthen the authors’ arguments for effective data interchange. Please refer the revised Introduction for details.

LINE-BY-LINE COMMENTS

Line 26: “Internet” is a proper noun and should be capitalized.

R: Correction was made as suggested.

Line 38: It is worth noting that data transmission of TDR and related instrument data from a bridge to an Internet-accessible database for near-real time interpretation dates back at least as early as 1999. I recall work by Marron, Dowding, O’Connor, and others on bridges in California and Indiana, USA, during that era. Dowding’s lab put considerable effort into developing some early data interchange models for TDR and related instrument data in the mid-2000s; some of this work was summarized in thesis by Dussud in 2001 and by articles by Kosnik et al in the proceedings of Field Measurements in Geomechanics 2007 (ASCE Geotechnical Special Publication 175). Thus, while the authors have correctly stated that TDR analysis is complex and that well-considered models for interoperability are useful and necessary, it is not accurate to say that no interoperable models existed before this current work. (I do agree that the authors’ work is novel and innovative – it’s just not the first effort in this area).

 R: Authors appreciate the comments and we already revealed the relevant references as suggested.

Line 44: “Hub” is capitalized as a proper noun. Is this a trade name, standard, or similar? If so, please define and provide a reference.

R: Authors appreciate the comments, and the details were made to clearly state that “… the hub approach, which is achieved by implementing different connectors corresponding to different Sensor Web and Internet of Things (SW-IoT) ecosystems, needs to use customized connectors for different protocols, leading the significant development cost.”

Line 47: Referring to something (open standards, in this case) as the “ultimate” approach smacks of hyperbole. Please use a less loaded term, such as “attractive” or “useful”.

R: Correction was made as suggested.

Line 50: Similarly, unless it can be quantitatively proven, please avoid phrases such as “the most comprehensive”. “..as a comprehensive solution” would be sufficient.

R: Correction was made as suggested.

Line 64: I recommend moving the introduction of XML to about Line 52 (since SensorML and WaterML are XML derivatives, it is useful to discuss XML in general before mentioning SensorML, ec)

R: Authors appreciate the comments, and the introduction of the XML is revealed as suggested.

Line 67: Please define “REST” here – not all readers will be familiar with how HTTP and architectures built upon it work. It is mentioned again on Line 71 few lines later, and again much later in the document in terms of a RESTful API – please provide a very brief definition and link to a reference document (I think it was defined in a dissertation circa 2000). It would probably be sufficient to say that REST provides a means for stateless information exchange over HTTP for Web services.

 R: Authors appreciate the comments, and the brief introduction of the REST is revealed as suggested.

Line 71: Provide a reference to the JSON standard or reference implementation if possible

 R: A reference to the JSON standard was added as suggested.

Line 74: “It is believed….” Believed by whom?  “…that XML represents the most basic”  Again, “the most” should be omitted unless it can be demonstrated quantitatively.

 R: Authors appreciate the comment, and the correction was made as suggested.

Line 85: Consider mentioning a popular concise definition of TDR: “cable radar” or “radar along a cable”

 R: Authors appreciate the comment, and the correction was made as suggested.

Line 96: Consider replacing “only a common” with “an off-the-shelf” coaxial transmission line. Strike “and mainly” and replace with “commonly”.

 R: The correction was made as suggested.

Line 105: it would be more accurate to refer to the failure surface as a “shear surface” rather than “fracture layer”

 R: The correction was made as suggested.

Line 115: The manufacturer of the TDR100/200 units is listed; can you list a manufacturer for the TDR3000/SYMPULS?

 R: The manufacturer for the TDR3000 is provided with Sympuls Aachen

Line 164: please define “Schematron”

 R: Schematron is known as a structural based validation language, and it is as an alternative to existing grammar based approaches

Line 169: “Restful” should be written “RESTful” to refer to the HTTP architecture.

 R: The correction was made as suggested.

Line 195: “There is no limit to its description”. No limit in what sense? Size of the data structure? Please clarify.

 R: This is revised as “There is no limit in determining the observed target to its description.”

Line 198: Consider phrasing “represents none to multiple” as “represents zero or more” to align with common usage in UML cardinality descriptions

 R: The correction was made as suggested.

Line 201: What is meant by “considered as a phenomenon”?

 R: The sentence is revised as “The observedProperty is considered as the property or phenomenon whose value is described or estimated through observation, and must be related to the featureOfInterest.”

Line 204: Consider phrasing “cannot be repeated” as “must be unique”. In general, it is better to state what something IS rather than what it IS NOT.

 R: The correction was made as suggested.

Figure 4: This figure is difficult to read when printed in grayscale. Please consider removing the color backgrounds from the UML class rectangles. At a minimum, use a different color scheme than red text on a pink background for the parameters of the central OM_Observation class.

 R: The correction was made as suggested, please refer to new Figure 4.

Line 212: Please provide a brief (one sentence) description of the SOAP protocol, and link to a specification or reference document. Strike “KVP”, as this abbreviation is never used again.

 R: The correction was made as suggested.

Line 239: Is the SOS “Degree” or “Deegree”? Spelling is inconsistent between the narrative and the reference list.  Also, the sentence seems to suggest that “based on Java” and “open sourced” are mutually exclusive, which they are not. Please clarify. Do you mean to suggest that the former is proprietary?  If so, the language choice is irrelevant.

 R: Authors appreciate the comment. First, Deegree SOS is confirmed. Second, to avoid the misunderstanding, the narrative is rewritten as “Among these, limited SOS functions (GetCapabilities, DescribeSensor, and GetObservation) are provided in Map Server SOS and Deegree SOS, while the 52°North SOS supports all SOS features including Transactional and Enhanced Operations. Thus, this study applies the 52°North SOS in the experiments.”

Line 245: “water level” and “deformation” are not proper nouns and thus should be all lowercase.

 R: The correction was made as suggested.

Line 261: Is “Measure Property” a noun or a verb (property or method)?

 R: The item is revised as “Measured Property”, as well as Figure 5.

Line 264: Strike “and cannot be repeated”. It is sufficient to say “unique”.

 R: The correction was made as suggested.

Figure 6 and elsewhere: please check for consistency of spelling and orthography for class/property/method names between the UML diagrams and the narrative text (e.g. TDR_WL_Observation in the diagram vs TDR_WL_Obsrvation in the narrative).

 R: Authors appreciate the comment, and the consistency check was proceeded as suggested.

Figure 9, Section 4.3., and related sections: “Part A”, “Part B”, and so on should be capitalized. Also, consider adding line numbers to code listings and refer to the individual parts by line number.

 R: Authors appreciate the comment, and we decide use adding line numbers to code listings and refer to the individual parts by line number.

Line 349: The figure cross reference here is broken.

 R: The correction was made as suggested.

Line 358: As the abbreviation “PPM” is never used again, it is sufficient to write “parts per million” (lowercase)

 R: The correction was made as suggested.

Line 360: How are features that are not points in plan view (e.g. a directionally drilled TDR cable) represented by latitude/longitude?

 R: Author appreciate the comment. Yes, now the information of latitude/longitude is appropriate for a point feature in plan view. For the case such as a directionally drilled TDR cable, the coordinates of the control points of the cable are suggested be provided sequentially to reveal the line feature in space

Figure 8 / Line 366: why is this line highlighted?

 R: The highlighted line was removed as suggested.

Figure 10: why is a line in Part B highlighted?

 R: The highlighted line was removed as suggested.

Figure 11: why is a line near the bottom highlighted?

 R: The highlighted line was removed as suggested.

Figure 12 is blurry and is too small to be legible.

 R: The resolution of Figure 12 is increased as suggested.

Line 400: “…of this observation, which should have been registered….” Is there an exceptional case where the Procedure would not have been registered?  If not, it is sufficient to write “…this observation, which was measured….”

 R: The correction was made as suggested.

Line 406: This is the first mention of GML. Please define it and provide a citation.

 R: The correction was made as suggested.

Line 429: “52o North SOS” – shouldn’t that be a degree sign, rather than a lowercase ‘o’?

 R: The correction was made as suggested.

Line 430 (and again on 443): “Published Configuration Database” is capitalized as though it were a proper noun. Is this a particular information model object? If so, where was it defined?  Please clarify.

 R: We changed the “Published Configuration Database” in lowercase.

Line 452: “adjust the threshold range by multiplying the standard deviation to the TDR waveform”. The standard deviation of what? A “noise” signal? Please clarify.

 R: The standard deviation is obtained from the differential result between measured waveform and the reference waveform. Thus the related description was made as suggested.

Line 475: Consider replacing the word “accurately” with “unambiguously” or similar to avoid semantic confusion with “accuracy” in the metrology sense.

R: The correction was made as suggested.

Reviewer 2 Report

A very useful model for TDR measurement data, which was introduced. However TDR measurements are not precise or not enough indicative, advantages are in way of installation process and costs. Therefore this technology is widely used and some standards are appreciated. Many authors partially resolved problems with noise and heterogeneity of waveforms. Maybe some reference citations can be added, but key topic is about standard measurement model.

Some formal errors and sentence composition should be improved:

P2, r. 44 This heterogeneity issue  is  usually solved “Hub solutions”  describe what is it Hub solution

P2. R.83 Today's TDR technology is used widely in geological materials?

P3. R.95 The sensing waveguide is only a common coaxial transmission line and mainly placed into a borehole and grouted in the soil or rock slope to be monitored. – rewrite the sentence

p.4 Figure 2 text TDR soil moisture penetrometer

5 r. 169 The standard can be extended be implemented with Restful transmission 8 Figure 5 text inside Project Info

p.18 r. 480 demonstrations are stored

20 r. 482 O’Connor

Improve the resolution of Figure 12, 13 and 15.

Author Response

Responses of Reviewer 2 comments:

A very useful model for TDR measurement data, which was introduced. However TDR measurements are not precise or not enough indicative, advantages are in way of installation process and costs. Therefore this technology is widely used and some standards are appreciated. Many authors partially resolved problems with noise and heterogeneity of waveforms. Maybe some reference citations can be added, but key topic is about standard measurement model.

R: Authors appreciate the comments. Following please find the response to each comments in detail.

Some formal errors and sentence composition should be improved:

P2, r. 44 This heterogeneity issue is usually solved “Hub solutions” describe what is it Hub solution

 R: Authors appreciate the comments, and the details were made to clearly state that “… the hub approach, which is achieved by implementing different connectors corresponding to different Sensor Web and Internet of Things (SW-IoT) ecosystems, needs to use customized connectors for different protocols, leading the significant development cost.”

R.83 Today's TDR technology is used widely in geological materials?

 R: “geological materials” is revised as “geotechnical engineering”.

R.95 The sensing waveguide is only a common coaxial transmission line and mainly placed into a borehole and grouted in the soil or rock slope to be monitored. – rewrite the sentence

 R: The sentence is rewritten as “The sensing waveguide is an off-the-shelf coaxial transmission line and mainly placed into a borehole with the grout in a soil/rock slope.”

4 Figure 2 text TDR soil moisture penetrometer

 R: The text “TDR soil moisture penetrometer” is corrected in Figure 2 as suggested.

5 r. 169 The standard can be extended be implemented with Restful transmission

 R: The text “RESTful” is suggested keep to refer to the HTTP architecture as commented by Reviewer 1.

8 Figure 5 text inside Project Info

 R: The text “ProjectInfo” is corrected as “Project Info”

18 r. 460 demonstrations are stored

 R: The correction was made as suggested.

20 r. 482 O’Connor

 R: The correction was made as suggested.

Improve the resolution of Figure 12, 13 and 15.

 R: Authors appreciate the comments. For the consistency requirement, Figure 8~17 were presented with the code listings as text in a dedicated listing environment (with line numbers) to increase the resolution.

Reviewer 3 Report

The authors describe a representation for a data exchange format for their TDR sensor experiment using OGC SWE components, SOS, SensorML and SWE Commons. The article is well structured, exhibits the technical and scientific depth in the general descriptive methods and implementations sections as one would expect. However, a discussion and comparison with existing literature and implementations is completely missing. There is no evaluation or criteria selected or documented that their implementation is correct/good and that their choice of encoding the SensorML profile for their TDR sensor could be done better or differently. I believe this is a bit of a shortcoming of this otherwise promising article. I would really suggest another section that discusses pros and also cons of the chosen approaches. Additionally, I’d suggest another proof-reading, as still some grammatical and spelling inconsistencies exist.

(Chung et al., under review) citation shows often, I am a bit skeptical, in the references section

Language-wise I would make following suggestions:

43 Hub solutions that communicate with

44 different ecosystems via customized connectors [9]. However, the Hub approach needs lots of

45 resource and effort to correctly understand and implement connectors for different protocols.

Why “the Hub” upper case? And “needs lots of resource and effort” is a bit too casual and vague

58 one of “the” standard information models

60 OGC standards. It attempts being … the two separate sentences should be combined. They make no sense stand-alone.

239 Degree SOS → Deegree SOS

Author Response

Responses of Reviewer 3 comments:

The authors describe a representation for a data exchange format for their TDR sensor experiment using OGC SWE components, SOS, SensorML and SWE Commons. The article is well structured, exhibits the technical and scientific depth in the general descriptive methods and implementations sections as one would expect. However, a discussion and comparison with existing literature and implementations is completely missing. There is no evaluation or criteria selected or documented that their implementation is correct/good and that their choice of encoding the SensorML profile for their TDR sensor could be done better or differently. I believe this is a bit of a shortcoming of this otherwise promising article. I would really suggest another section that discusses pros and also cons of the chosen approaches. Additionally, I’d suggest another proof-reading, as still some grammatical and spelling inconsistencies exist.

R: Authors appreciate the comments.

Previous works have put considerable effort into developing some early data interchange models for TDR and related instrument data [25,26]; However, the current TDR analysis with the related parameters are still complex and that well-considered models for interoperability are useful and necessary. Thus, this study aims to improve the interoperability of TDR sensor metadata and observations by proposing a TDR data model profile based on the OGC Observations and Measurements (O&M) international open standard. Authors agree that there is no evaluation or criteria selected or documented that current implementation is correct/good and the choice of encoding the SensorML profile for TDR sensor could be done better or differently. But authors believe that this study provides the very first step by proposing the interoperability of TDR sensor metadata and observations based on the revealed successful works of Pearlman et al. [50] and Sofos et al. [33], and this development would be a reference for the next phase of the related research to have a further comparison using such as SensorThing API in near future. Another proof-reading is done for grammatical and spelling inconsistencies as suggested.

(Chung et al., under review) citation shows often, I am a bit skeptical, in the references section

R: Since the analysis procedures of TDR landslide monitoring proposed by Chung et al., (under review) citation is one of the basics in which is applied in the study, authors made effort to keep the reference for the possible publication in near future.

Language-wise I would make following suggestions:

43 Hub solutions that communicate with different ecosystems via customized connectors [9]. However, the Hub approach needs lots of resource and effort to correctly understand and implement connectors for different protocols.

Why “the Hub” upper case? And “needs lots of resource and effort” is a bit too casual and vague

R: Authors appreciate the comments, and the details were made to clearly state that “… the hub approach, which is achieved by implementing different connectors corresponding to different Sensor Web and Internet of Things (SW-IoT) ecosystems, needs to use customized connectors for different protocols, leading the significant development cost.”

58 one of “the” standard information models

       R: The correction was made as suggested.

60 OGC standards. It attempts being … the two separate sentences should be combined. They make no sense stand-alone.

R: The correction was made as suggested.

239 Degree SOS → Deegree SOS

R: The correction was made as suggested.

Round 2

Reviewer 1 Report

I am satisfied with the authors' responses to my review comments and thank them for the quick turnaround.

A few minor things that should be corrected before publications:

Line 30:  "electromagnet (EM) wave" should be "electromagnetic (EM wave)"

Lines 98-99: Recommend rephrasing as "TDR is commonly known as "cable radar", as the operating principle of TDR is similar to that of radar, except that the EM pulse travels through a cable rather than free space." or similar

Line 228 still has a broken reference (after reference 44). Please fix or delete.

Line 348: ObservableProperty is misspelled

Lines 408-409: FeatureOfInterest and Identifier as misspelled

Line 562 / Reference 26: The publication year is shown incorrectly. The 7th Int'l Symposium on Field Measurements in Geomechanics was held in 2007, not 2012 (evidently the proceedings were published online in 2012, but that is not the date to be listed in the reference).

Author Response

Responses of Reviewer 1 comments:

I am satisfied with the authors' responses to my review comments and thank them for the quick turnaround.

A few minor things that should be corrected before publications:

R: Authors appreciate the comments. Following please find the responses to each comments in detail.

Line 30:  "electromagnet (EM) wave" should be "electromagnetic (EM wave)"

R: Authors appreciate the comments. Correction has been made as suggested.

Lines 98-99: Recommend rephrasing as "TDR is commonly known as "cable radar", as the operating principle of TDR is similar to that of radar, except that the EM pulse travels through a cable rather than free space." or similar

R: Authors appreciate the comments. Correction has been made as suggested.

Line 228 still has a broken reference (after reference 44). Please fix or delete.

R: Authors appreciate the comments. The original manuscript is correct to reveal the reference, however, it has a broken reference after transforming to PDF. Correction has been made as suggested.

Line 348: ObservableProperty is misspelled

R: Authors appreciate the comments. Correction of “ObservableProperty” has been made as suggested.

Lines 408-409: FeatureOfInterest and Identifier as misspelled

R: Authors appreciate the comments. Correction has been made as suggested.

Line 562 / Reference 26: The publication year is shown incorrectly. The 7th Int'l Symposium on Field Measurements in Geomechanics was held in 2007, not 2012 (evidently the proceedings were published online in 2012, but that is not the date to be listed in the reference).

R: Authors appreciate the comments. Correction has been made as suggested.

Reviewer 3 Report

ll.53-54: leading the significant development cost. -> leading "to" significant development cost.

ll.61-62: ... various types of sensor metadata [30] based on Extensible Markup Language (XML) derivative -> based on "an" Extensible Markup Language (XML) derivative. / also the citation could be at the end of this sentence.

ll.77-78: ... Web services that conform to the REST architectural style are called RESTful Web services (RWS) and providing interoperability. are called ... and provide, I believe the providing form here is grammatically wrong.

ll.167: SensorML, which is one of several implementation standards produced under OGC’s SWE activity as aforementioned, much focus on the process of measurement and observation, rather than on sensor hardware. / Thissentence is grammatically not connected, in particular the middle part, much focus - what?

ll.169: ... on sensor hardware. Thus"," SensorML / comma required

ll.173: The standard is already provided to specify models and an XML implementation for the SensorML V2.0. / The standard "is" provided? .. for "the" SensorML? this doesn't make sense.

etc. etc.

Overall the technical content of the article was sharpened, but the textual quality is still sub par.

That the authors state in the response that "Another proof-reading is done for grammatical and spelling inconsistencies as suggested. ", does not seem credible. It really needs proper English editing.

Beyond that I I believe the paper could be published. I am not an English editor for the paper.

Author Response

Responses of Reviewer 3 comments:

ll.53-54: leading the significant development cost. -> leading "to" significant development cost.

R: Authors appreciate the comments. Correction has been made as suggested.

ll.61-62: ... various types of sensor metadata [30] based on Extensible Markup Language (XML) derivative -> based on "an" Extensible Markup Language (XML) derivative. / also the citation could be at the end of this sentence.

R: Authors appreciate the comments. Correction has been made as suggested.

ll.77-78: ... Web services that conform to the REST architectural style are called RESTful Web services (RWS) and providing interoperability. are called ... and provide, I believe the providing form here is grammatically wrong.

R: Authors appreciate the comments. Correction has been made as suggested.

ll.167: SensorML, which is one of several implementation standards produced under OGC’s SWE activity as aforementioned, much focus on the process of measurement and observation, rather than on sensor hardware. / This sentence is grammatically not connected, in particular the middle part, much focus - what?

R: Authors appreciate the comments. Correction has been made as “SensorML, which is one of several implementation standards produced under OGC’s SWE activity as aforementioned, focuses on the process of measurement and observation…”

ll.169: ... on sensor hardware. Thus"," SensorML / comma required

R: Authors appreciate the comments. Correction has been made as suggested.

ll.173: The standard is already provided to specify models and an XML implementation for the SensorML V2.0. / The standard "is" provided? .. for "the" SensorML? this doesn't make sense.

R: Authors appreciate the comments. Correction has been made as “The standard of SensorML V2.0 proposes sensor metadata models and an XML implementation.”

etc. etc.

Overall the technical content of the article was sharpened, but the textual quality is still sub par. That the authors state in the response that "Another proof-reading is done for grammatical and spelling inconsistencies as suggested. ", does not seem credible. It really needs proper English editing.

Beyond that I I believe the paper could be published. I am not an English editor for the paper.

R: Authors appreciate the comments. Proper English editing is requested for grammatical and spelling inconsistencies as suggested.
